# Efficient Spatial Sampling for AFM-Based Cancer Diagnostics: A Comparison between Neural Networks and Conventional Data Analysis

**Gabriele Ciasca** [1,2,*], **Alberto Mazzini** [1,2] **, Tanya E. Sassun** [3]**, Matteo Nardini** [1,2]**, Eleonora Minelli** [1,2]**, Massimiliano Papi** [1,2]**, Valentina Palmieri** [1,2] **and Marco de Spirito** [1,2]

[1] Physics Institute, Catholic University of Sacred Heart, Largo F. Vito, 1 00168 Rome, Italy;
alberto.mazzini01@icatt.it (A.M.); matt.nardini@gmail.com (M.N.); eleonora.minelli@unicatt.it (E.M.);
massimiliano.papi@unicatt.it (M.P.); valentina.palmieri@unicatt.it (V.P.); marco.despirito@unicatt.it (M.d.S.)

[2] Fondazione Policlinico Universitario Agostino Gemelli IRCCS, 1 00168 Rome, Italy

[3] Department of Neurology and Psychiatry, Division of Neurosurgery, Policlinico Umberto I, Sapienza University of Rome, Viale del Policlinico, 155 00161 Rome, Italy; tanya.sassun@gmail.com

* Correspondence: gabriele.ciasca@unicatt.it

**Abstract:** Atomic force microscopy (AFM) in spectroscopy mode receives a lot of attention because of its potential in distinguishing between healthy and cancer tissues. However, the AFM translational process in clinical practice is hindered by the fact that it is a time-consuming technique in terms of measurement and analysis time. In this paper, we attempt to address both issues. We propose the use of neural networks for pattern recognition to automatically classify AFM force–distance (FD) curves, with the aim of avoiding curve-fitting with the Sneddon model or more complicated ones. We investigated the applicability of this method to the classification of brain cancer tissues. The performance of the classifier was evaluated with receiving operating characteristic (ROC) curves for the approach and retract curves separately and in combination with each other. Although more complex and comprehensive models are required to demonstrate the general applicability of the proposed approach, preliminary evidence is given for the accuracy of the results, and arguments are presented to support the possible applicability of neural networks to the classification of brain cancer tissues. Moreover, we propose a possible strategy to shorten measurement times based on the estimation of the minimum number of FD curves needed to classify a tissue with a confidence level of 0.005. Taken together, these results have the potential to stimulate the design of more effective protocols to reduce AFM measurement times and to get rid of curve-fitting, which is a complex and time-consuming issue that requires experienced staff with a strong data-analysis background.

**Keywords:** atomic force microscopy; biomechanics; brain cancer

---

## 1. Introduction

As a matter of fact, tissues grow and remodel themselves in response to various mechanical and physical forces [1–8]. In physiological conditions, these mechanical cues influence biochemical reactions in cells, modulating cellular processes such as proliferation, differentiation, and apoptosis, which are crucial for organ development and maintenance. The way cells sense forces is largely mediated by the extracellular matrix (ECM), which is an ensemble of extracellular macromolecules which provide structural and biochemical support for surrounding cells [3,8,9]. ECM composition and morphology largely depend on its location in organs and tissues, and its properties change in time due to physiological (e.g., aging [10–12]) and pathological conditions (e.g., cancer [2,5,7,13–21]).

Healthy tissues are characterized by a mechanical homeostasis, in which forces among cells and their ECM undergo a dynamical balance. The onset and the development of a wide range of pathologies is accompanied by a disruption of this physical homeostasis [7,22–25]. Cancer is one of the diseases that is most affected by such mechanical alterations [4–6,9,13,16–19,21,25–30]. Neoplastic cells are usually softer than their healthy counterparts, showing also changes in shape and adhesion properties that are associated with high proliferation rates, loss of differentiation, escape from apoptosis, and high spreading capability in the case of metastasis [5,31–33]. Conversely, cancer tissues are usually stiffer than healthy ones, as a direct consequence of an increased deposition of ECM fibrous proteins, such as collagen, fibronectin, and laminin [13,15,16,21]. This increase in stiffness is a fundamental hallmark of the pathology and is also one of the main reasons why palpation remains a key tool for the diagnosis of some tumors.

Mechanical and structural modifications of the samples can be studied by atomic force microscopy (AFM) in force spectroscopy mode and imaging mode [15–18,21,22,28–30,34–38]. In a force spectroscopy experiment, AFM collects the so-called force–distance (FD) curves, a plot of the AFM cantilever deflection as a function of the tip-sample separation. These curves can be mathematically analyzed to obtain quantitative information about the elastic, viscous, and adhesive behavior of samples. In this context, one of the most used parameters is Young's modulus (E), which provides information on the sample stiffness. Lekka and co-workers performed the first application of this technique to tissue biopsies for cancer diagnosis and staging, showing that the AFM Young's modulus allows for the detection of cancer cells in tissue slices [30]. Plodinec et al. demonstrated that stiffness maps of human breast biopsies could help in assessing cancer stages [15]. Tian and collaborators performed indentation-type AFM experiments on liver cancer tissues, revealing that nanoscale tissue elasticity can help physicians grade pathological states of the liver, distinguishing among normal livers and those suffering from cirrhosis, hepatocellular carcinoma, and its recurrence [16]. Similarly, we recently unveiled the nano-mechanical signature of malignant and benign brain cancer tissues [21].

All these results substantially agree with the fact that AFM has the capability of distinguishing between healthy and cancerous tissues, thus opening wide possibilities for cancer diagnosis and staging. In this context, AFM could be used in diagnostics in combination with standard methods, such as histological section analysis. It is worth stressing that AFM has also several advantages over these methodologies, because this technique yields quantitative results which can be statistically analyzed [15,19,28,30,39–41].

Despite this potential, the AFM translational process into clinical practice is still hindered by the fact that it is a time-consuming technique in terms of both measurement and analysis [29]. In this paper, we attempt to address both issues. As far as the problem of analysis times is concerned, we propose the use of neural networks to classify AFM force–distance (FD) curves in an automated fashion. We tested this method on a clinically relevant problem, i.e., discriminating different types of surgically removed brain cancer tissues obtained from patients diagnosed with glioblastoma multiforme, one of the most frequent malignant brain cancers, which is characterized by a highly aggressive behavior and an unfavorable prognosis [42]. A second bottleneck of the AFM technique, which is a limitation of its use in medical practice as a diagnostic tool, is the long time cost of measurements. Here, we discuss a possible strategy to shorten this time based on the estimation of the minimum number of FD curves needed to classify a tissue with a 0.005 confidence level, modeling AFM mapping in terms of a binomial process. These statistical considerations have the potential to boost the translational process of AFM in clinical practice, because they help define an optimum spatial sampling for FD curves, which allows reducing experimental time, while simultaneously limiting the loss of accuracy.

## 2. Materials and Methods

### 2.1. Patients Recruitment and Sample Preparation

Fifteen patients with brain cancer, eight of which were diagnosed with glioblastoma multiforme (GBM), were recruited for the study. Recruitment was carried out with informed consent and was approved by the institutional review board (Ref. no. 3508 Prot. 1706/15). Human glioblastoma tissues were obtained after surgical resection and underwent AFM analysis within two hours of removal. Histological examinations of tissues were systematically compared with AFM outcomes to assess the presence and the stage of cancer. The direct comparison between FD curves and histological findings for the data analyzed in the present paper was deeply discussed in Reference [21]. Briefly, immediately after surgery, brain tissues were divided into classes by expert neurosurgeons and pathologists, considering also the results of radiological examinations. The same samples were divided in two parts, one of which underwent AFM analysis with the remaining part used for histological findings. Different staining techniques were used; specifically, (i) hematoxylin and eosin staining was used to classify surgically removed tissues into healthy peri-tumoral tissues, necrotic tissues, and tumor non-necrotic tissues; (ii) Gomori staining was used to assess the overexpression of fibrous proteins, which in turn was correlated with the stiffening of cancer tissues as quantified by means of Young's modulus E; (iii) Alcian-blue staining was used to assess the presence of an increased expression of hyaluronic acid, which was correlated with an alteration of tissue viscosity. Further and detailed information on the histological examination of the sample examined in this study can be found in Reference [21].

### 2.2. AFM Measurements

Immediately after tumor resection, tissue samples were cut in sections with a surgical scalpel. Specimens were sampled in different positions within the solid tumor, and then immobilized in a Petri dish with a thin layer of fast-drying dual epoxy bio-compatible glue according to Reference [15]. The presence and the relevance of possible artefacts induced by glue infiltration was discussed in Reference [21]. All the preliminary steps were performed in a ringer buffer. AFM measurements were performed in a liquid and at room temperature using a JPK NanoWizard-II microscope coupled with a ZEISS Axio-Observer Fluorescence Inverted Microscope. GBM tissues were investigated using Si cantilevers (Mikromash, HQ:CSC38/NO AL) with a spring constant of ~0.05 N/m, which was determined before each measurement by thermal calibration. An indentation force of 5 nN and a velocity during indentation of 5 μm/s were adopted. Indentation forces of the same order were used in Reference [15] for breast cancer tissues and Reference [16] for hepatocellular carcinoma. The problem of the extreme surface corrugation of postoperative tissues was overcome using an additional piezoelectric actuator with a *z*-range of 100 μm [21,37].

### 2.3. AFM Data Analysis

Two types of data analysis were performed. Firstly, we applied the conventional analysis based on FD curve-fitting to measure the apparent Young's modulus E of tissues. Then, we investigated the feasibility of automated FD curve classification by means of a neural network for pattern recognition.

The apparent Young's modulus E (referred to in the following as Young's modulus E) was retrieved by fitting the Sneddon model for a conical indenter to each FD curve with the JPK data processing software as follows:

$$F(\delta) = \frac{2\,E\,\tan(\alpha)}{\pi(1-v^2)}\,\delta^2, \tag{1}$$

where $\alpha$ accounts for the half-aperture angle of the conic tip (equal to 20° for the exploited tips), $v$ is the Poisson ratio (set at 0.5 considering the material incompressibility), and $\delta$ is the indentation depth. Moreover, the work of adhesion (W) was measured as the area between the approach and retract curves; W represents the amount of energy dissipated when the tip is completely detached from the sample.

Because biological samples, as well as complex materials, are characterized by spatially heterogeneity, their physical and functional properties are better described by nano-mapping rather than randomly located single-point measurements [15,43–50]. Therefore, in order to take into account the spatial distribution of E, we acquired stiffness maps of the sample with a typical size of 40 μm × 40 μm; map resolutions ranged from 1.25 to 5 μm.

Secondly, we applied a pattern recognition neural network-based algorithm, which classifies FD curves with the aim of avoiding curve-fitting. The algorithm was implemented in Matlab [29,50,51]. For the learning phase, we exploited the Levenberg–Marquardt backpropagation algorithm and a set of 200 FD input profiles whose class memberships were already assigned. The choice of 200 FD profiles is further commented on in Figure A1. Input profiles were randomly extracted from all the recruited patients. The selected input profiles were randomly divided into three sets: 70% were used for training, 15% were used to avoid overfitting, and the last 15% were used as an independent test of network generalization. During the training phase, internal weights of the network were adjusted to minimize differences between the expected outputs and the network outputs. After the training phase, the network was exploited to classify unknown experimental FD curves using the feed-forward neural network (FFNN) algorithm. The FFNN algorithm is composed of three layers: the input, the hidden, and the output layers, each of which has a different neuron number. Prior to the feed of the NN with FD curves, we interpolated them in such a way that each curve was composed of 200 points with the same $x$-coordinates. Consequently, we chose 200 neurons in the input layer. The number of neurons in the hidden layer was optimized to minimize computational cost and maximize the training performance, as well as the correct outputs. We chose to test the performance of this method in distinguishing between two classes of tissues at a time.

## 2.4. Statistics

Statistical analyses were performed using the software package R (3.5.2 release) [52]. E values were reported as mean ± standard error of the mean (SEM). E distributions were analyzed with the modified Levene equal variance test (F-test) to determine whether variances in the three classes of tissues (necrotic and non-necrotic tumor tissues and healthy tissues) were equal. E values were tested for normality by a visual inspection of the quantile–quantile (QQ) plot followed by a Shapiro–Wilk test. Welch's ANOVA, along with a Games–Howell post hoc analysis, was used to assess the presence of statistically significant differences among the three classes of tissues [53].

The performance of the neural network and Young's modulus E in discriminating between (i) non-necrotic GBM and healthy tissues, (ii) necrotic GBM and healthy tissues, and (iii) necrotic and non-necrotic GBM tissues was evaluated by computing the receiver operating characteristic (ROC) curve [54]. To this purpose, NN and E outputs were compared to our a priori knowledge based on the analysis of histological sections. ROC analysis is a widely used technique in the medical field for evaluating diagnostic tools. It is a two-dimensional graph in which the true positive rate, $tp$ or sensitivity, is plotted against the false positive rate, $fp$ or $1 - $ specificity. Mathematically, $tp = TP/(TP + FN)$ and $fp = FP/(FP + TN)$, where $TP$ indicates a true positive that occurs when a positive instance is classified as positive; $FN$ indicates a false negative, which occurs when a positive instance is classified as negative; $TN$ indicates a true negative, i.e., a negative instance that is classified as negative; $FP$ is a false positive, i.e., a negative instance that is classified as negative.

ROC curves were calculated using the R package pROC [55]. A logistic regression was executed in order to combine the ROC curves calculated from the approach and retract curves separately with the neural network approach. For this aim, we used the function $glm()$ from the R package stats [52]. For our purposes, it was useful to calculate the accuracy, referred to as $p$ in the following, which is an additional statistical metric calculated for ROC curves. Accuracy is defined as $p = \frac{TP+TN}{TP+FN+TN+FP}$. For a continuous classifier, such as E, we can calculate a specific accuracy value for each couple of $tp$

and *fp* values, whereas *p* can be easily derived as a function of the $x = \frac{FP}{FP+TN}$ and $y = \frac{TP}{TP+FN}$ variables in the ROC curve, by means of Equation (2) [29].

$$p(x,y) = \frac{Ptot \cdot y + Ntot \cdot (1-x)}{Ptot + Ntot}, \tag{2}$$

where $P_{tot}$ and $N_{tot}$ are the total numbers of curves belonging to the positive and negative actual classes, respectively. For this study, a total of 10,240 FD curves, of which 4352 described necrotic tissues, 4352 described tumor tissues, and the remaining 1536 described healthy tissues, were analyzed using neural networks.

## 3. Results and Discussion

### 3.1. Classification of Brain Cancer Tissues on the Sneddon Model Basis

Glioblastoma multiforme (GBM) is the most common and aggressive type of malignant brain tumor, characterized by the presence of poorly differentiated neoplastic astrocytes, newly formed tumor micro-vessels, and necrotic regions. The microvasculature proliferation and the necrotic regions are specific hallmarks of the pathology [21,42,56,57]. These biochemical and morphological modifications of tumor microenvironment are also associated with a significant alteration of tissue biomechanics, which can be investigated by AFM through the acquisition of FD curves.

In Figure 1a–c, we show a set of representative FD approach curves measured on necrotic GBM tissues (a), healthy peri-tumoral tissues (b), and GBM cancer tissues (c), together with a smoothed curve of the entire reported dataset. FD curves were randomly selected from our database, according to the specific tissue type. FD curves acquired in different brain regions appear to be qualitatively different, unveiling an inhomogeneous mechanical response of the samples. A qualitative analysis of Figure 1 shows that necrotic tissues (a) are softer than healthy peri-tumoral tissues (b), which in turn are softer than cancer ones.

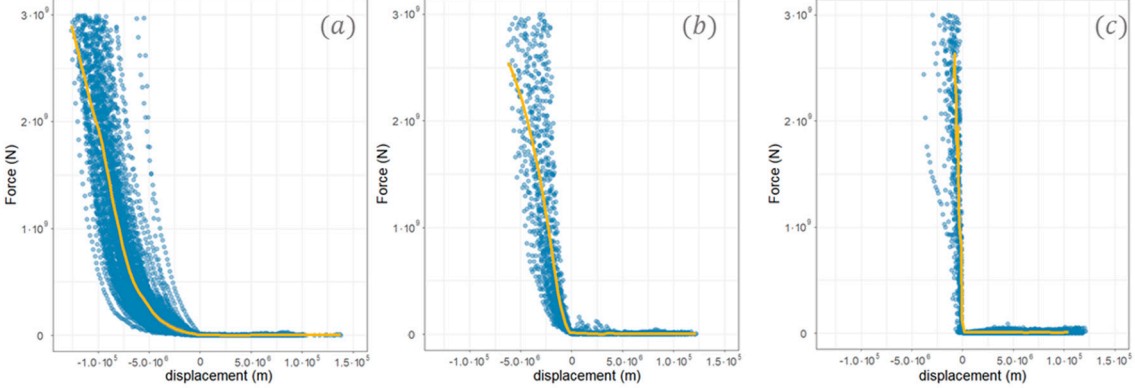

**Figure 1.** Representative force–distance profiles (approach curves) for necrotic glioblastoma multiforme (GBM) tissue (**a**), healthy tissues (**b**), and non-necrotic tumor tissues (**c**). A smoothed curve of the three datasets is reported as a continuous gold line.

FD approach curves can be analyzed with Equation (1) to obtain the apparent Young's modulus E, which gives quantitative information on the sample stiffness. A box-plot analysis of the E values calculated from the entire database of FD curves is shown in Figure 2a. Welch's ANOVA for unequal variances reveals that E values differ significantly in the three tissue types (F[2,760] = 711.19, $p < 2.2 \times 10^{-16}$. A Games–Howell post hoc analysis confirms that cancer tissues are stiffer than healthy tissues ($p < 1 \times 10^{-6}$), which in turn are stiffer than necrotic ones ($p < 1 \times 10^{-6}$). This result suggests that the Young's modulus E can be used as a promising biomarker for the classification of brain cancer tissues. We assessed the performance of the Young's modulus E as a brain cancer biomarker evaluating the receiver operator characteristic (ROC) curve. The ROC curve is a technique to visualize classifiers

based on their performance, widely used in the medical field to evaluate and compare diagnostic tools. As previously described in Section 2.4, ROC curves are two-dimensional graphs in which *sensitivity*, also referred to as true positive (TP) rate, is plotted against 1 − specificity, which is the false positive (FP) rate. As specificity ranges between 0 and 1, ROC curves are usually also reported in terms of sensitivity as a function of specificity, inverting the *x*-axis. In the present paper, the latter notation is used. Figure 2b shows the ROC plots for the classification of FD curves according to the corresponding E values measured on GBM necrotic and healthy tissues (gold continuous line), tumor and healthy tissues (orange continuous line), and necrotic and tumor tissues (brown continuous line). The diagonal line represents the $y = x$ curve, which is the expected performance for a completely random classifier. For all the datasets, ROC curves rapidly increase for low values of the *x*-axis, showing that the Young's modulus E is a brain cancer biomarker with good classification ability. A widely used statistical metric for the quantitative evaluation of a ROC curve is the so-called area under the curve (AUC).

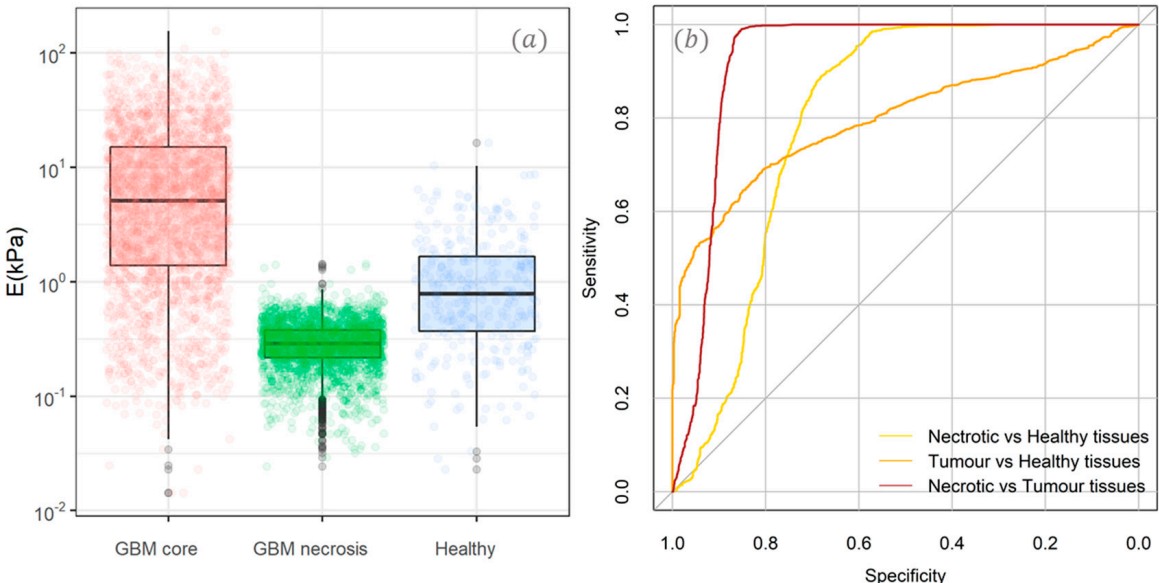

**Figure 2.** Box-plot analysis of the Young's modulus (E) values calculated for the three tissue types (**a**); receiver operator characteristic (ROC) plots for the classification of force–distance (FD) curves according to the corresponding E values (**b**). The diagonal line represents the expected performance for a completely random classifier.

By definition, AUC values lie in the range [0, 1], where 1 corresponds to an ideal classifier; in general, the higher the AUC value is, the better the classifier performance is. Large AUC values were measured for the three binary classification problems, i.e., 0.797 for healthy versus cancer tissues, 0.922 for necrotic versus healthy tissues, and 0.831 for necrotic versus tumor tissues.

Further information on the biomechanical response of tissues can be inferred from the analysis of the entire indentation cycle (Figure 3). This information includes AFM hysteresis (H), which represents percentual energy dissipated during indentation, and work of adhesion (W), which represents the amount of energy dissipated when the tip is completely detached from the sample. W is determined as described in the Section 2.3, and it is schematically represented in Figure 3. In Figure A2, we report the W box plot for the three different classes of tissues. We found no such marked differences as in the case of Young's modulus E, suggesting that the use of E is preferable for diagnostic purposes.

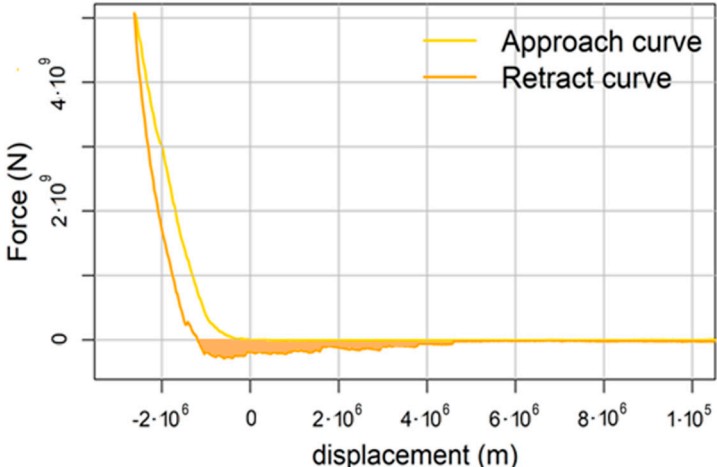

**Figure 3.** Representative force–distance cycle measured on a necrotic GBM tissue. Work adhesion is highlighted in orange.

### 3.2. Automated Classification of Brain Cancer Tissues Using a Neural Network Approach

Figure 2 suggests that AFM is a promising tool for brain cancer diagnosis and tissue classification. However, many factors still hinder the translational process of AFM in diagnostic practice. Among these factors, a central role is played by AFM curve-fitting, which is a complex, time-consuming issue that requires specialized personnel with a solid physical and mathematical background.

Since differences in mechanical properties translate into different shapes of FD curves, in a recent paper, we used neural networks for the automated classification of brain cancer tissues [29]. This method was applied to the same classification problems discussed in Figure 2b, with the aim of avoiding single-curve fitting. This approach has several advantages over the classical curve-fitting method. In particular, it is able to provide a direct classification of the measured tissues according to the tissue type, namely, healthy, necrotic, and non-necrotic tumor tissue. Moreover, while the Sneddon model takes into account only a small portion of the approach curve, neural networks for pattern recognition consider the entire force–distance cycle shown in Figure 3, including not only the approach curve (gold continuous line) but also the retract curve (orange continuous line), which is potentially a valuable source of clinical information. In a previous paper, we tested the performance of the neural network in discriminating among different tissue types, using the approach and retract curve separately. In this work, we combine the information coming from two curves in order to increase the effectiveness of the method. As described in the Section 2.4, we used a logistic analysis of the results to combine information from approach and retract curves in a single ROC plot.

In Figure 4, we show ROC curves for the classification of (a) GBM necrotic and non-necrotic tumor tissue, (b) healthy and tumor tissues, and (c) necrotic and healthy tissues. ROC plots calculated using the approach and the retract curves separately are shown as continuous gold and orange lines, respectively. ROC curves obtained combining the approach and retract profiles are shown as brown continuous lines. One can note that the combined ROC curves have a very large AUC, i.e., 0.895 for healthy versus cancer tissues, 0.988 for necrotic versus healthy tissues, and 0.987 for necrotic versus tumor tissues.

In Figure 5, we report a stem plot summarizing AUC values for the different classification approaches, namely those based on the Sneddon model and on the neural network. Taken together, the results in Figure 5 show that (i) the neural network approach exhibits a superior classification capability to the Sneddon model approach; (ii) the combination of the approach and retract curves by means of a logistic analysis improves the classification performances.

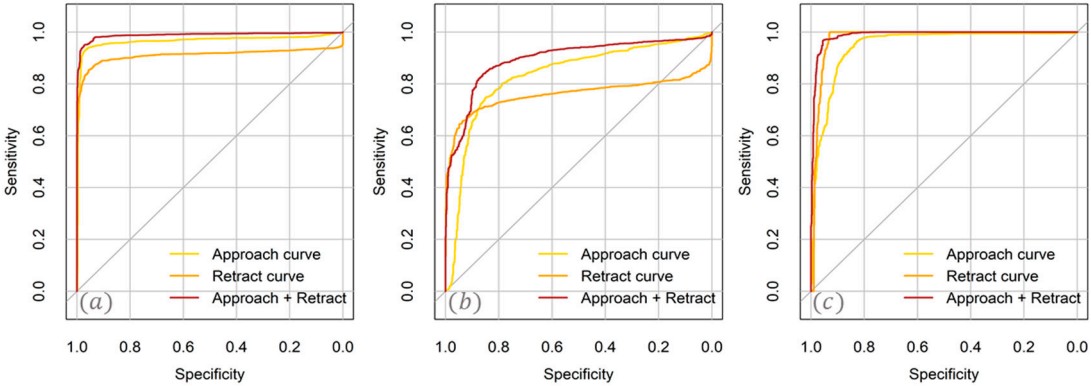

**Figure 4.** ROC curves for the classification of GBM necrotic and non-necrotic tumor tissues (**a**), healthy and tumor tissues (**b**), and necrotic and healthy tissues (**c**).

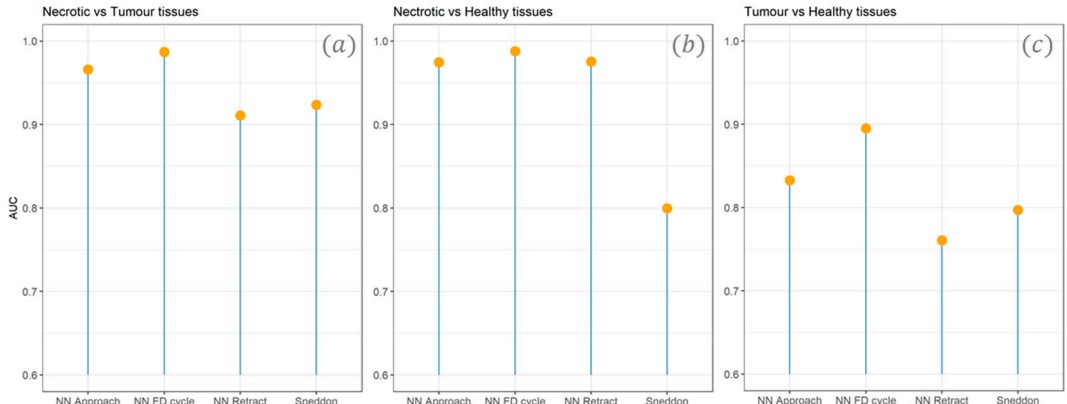

**Figure 5.** Stem plot with AUC values for the different explained classification approaches, namely those based on the Sneddon model and on the neural network (approach, FD cycle, and retract).

### 3.3. Determination of the Most Efficient AFM Sampling for the Classification of Brain Cancer Tissue

As stated above, the translational process of the AFM in cancer diagnostics is still hindered by the fact it is a time-consuming technique in terms of both measurement and analysis times. In the previous section, we showed that the use of neural network has the potential to help getting rid of the manual fitting for FD curves, thus dramatically reducing the time spent for data analysis.

In this section, we discuss a possible strategy to evaluate the minimum number of FD curves needed to properly classify a given scanning area, with a predetermined level of accuracy. Considering a recent seminal work in statistical methods for biomedical science, we chose a 0.005 confidence level, instead of the typical 0.05 level, which is associated with poor experimental reproducibility [58].

We started by trying to assess the class membership of a given tissue using a single-point measurement, acquired in a random location within a selected 40 μm × 40 μm area. For the sake of simplicity, let us assume that we have a tumor tissue and that we have two possible algorithm decisions: the correct one (tumor tissue) and the incorrect one (normal tissue).

The classification algorithms (either the one based on the Sneddon model (Figure 2) or the one based on neural networks (Figure 3)) will give us the proper diagnosis with a probability $p$ and the incorrect one with a probability $1 - p$, provided that we are making our decision on a single trial basis. This probability can be calculated directly from the ROC curve associated with the selected classification process. To clarify this point, it is useful to define the *accuracy*, a statistical metric that can be derived from the confusion matrix discussed in Section 2.4. Mathematically, accuracy is defined as $p = (TP + TN)/(TP + FN + TN + FP)$. This quantity represents the overall rate of success of the algorithm classification and, thus, we can associate it to the probability value we are

looking for. Similarly, $1 - p = (FP + FN)/(TP + FN + TN + FP)$ is the overall error rate of the classifier. The quantities of p and $1 - p$ can be calculated directly from the ROC curves of Figures 2 and 3 for the Sneddon model and the neural network model, respectively. Like other ROC-based statistics, accuracy depends on the selected threshold that, in turn, is associated with a couple of *x*- and *y*-values, where $x = 1 -$ specificity and $y =$ sensitivity. As discussed in Section 2.4, $p(x,y)$ can be calculated according to Equation (2). In Figure 6a, we show $p$ as a function of $1 -$ specificity for the ROC curves in Figure 2b, which evaluate the E performance in the classification of necrotic and healthy tissues (gold continuous line), tumor and healthy tissues (orange continuous line), and necrotic and tumor tissues (brown continuous line). Similar curves are shown in Figure 6b for the three ROC plots obtained by combining the approach and the retract curves. Provided that the threshold which maximizes accuracy is chosen, we can associate a probability of success $p$ to each classification process that is equal to the corresponding maximum value, $p_{max}$, in Figure 5. These maximum values are summarized in Table 1.

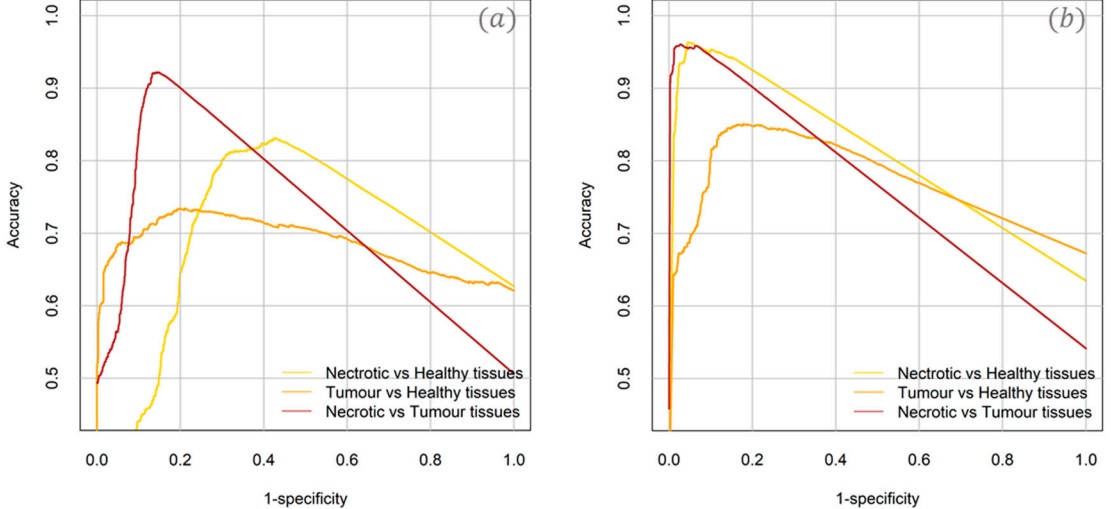

**Figure 6.** Accuracy $p$ as a function of $1 -$ specificity for the Young's modulus E (**a**) and the neural network (**b**).

**Table 1.** Minimum number of curves necessary to assign a correct outcome is shown, with the corresponding $p_{max}$ shown in brackets.

| Model | Healthy vs. Tumor Tissues | Necrotic vs. Tumor Tissues | Healthy vs. Necrotic Tissues |
|---|---|---|---|
| Sneddon model | 29 (0.73) | 7 (0.92) | 13 (0.83) |
| Neural Network | 13 (0.85) | 5 (0.96) | 5 (0.96) |

Once we assign the probability values, we can go back to our task, i.e., to assign the proper class membership to a tumor tissue when two algorithm decisions are possible (tumor and healthy tissues). Moreover, let us choose the neural network approach to solve this problem instead of the Sneddon model one. For this classification problem (tumor vs. normal tissues using neural networks), $p = 0.85$ (Table 1) and, thus, on a single curve basis, we would obtain a correct diagnosis in approximately 85% of the analyzed scanned areas and the wrong diagnosis in the remaining 15% of cases. As a matter of fact, probability of failure is far above the prefixed 5‰ level; therefore, we cannot make a decision using a single AFM FD curve. Thus, we increase the number of curves, which also increases measurement time, considering three FD curves instead of a single one. In this case, we have the following four possible outcomes: (i) each of the three FD curves is classified as a "tumor"; (ii) two out of three are classified as "tumor" and one out of three as "normal"; (iii) one out of three is classified as "tumor" and two as "normal"; (iv) all three points are classified as "normal". With these possible outcomes, assigning the class membership to the 40 μm × 40 μm scanned area is not straightforward, and it depends on the assignment strategy. We propose assigning the class membership to the predicted class that has the

majority of assignments. Using this method, in cases (i) and (ii), we get a correct diagnosis (tumor), while, in cases (iii) and (iv), we get the incorrect one (normal). It is easy to see that the probability of getting the proper diagnosis is equal to the probability of at least two successes; the probability of getting an incorrect diagnosis is equal to the probability of at least two failures. The calculation of such probabilities is straightforward if one considers that we have $n = 3$ independent trials (i.e., we can acquire three independent FD curves), trials are identical, and each trial has a given probability of success $p_{max}$ (which can be derived from Figure 5 under the aforementioned hypotheses) and a given probability of failure, $1 - p_{max}$. An experiment which satisfies this design is referred to as a binomial experiment and, thus, probabilities can be calculated using the formula $P(x) = C_n^x p_{max}^x \cdot (1 - p_{max})^{n-x}$, where $P(x)$ is the probability of having $x$ success with $n$ independent trials, $p_{max}$ is the probability of a success, $1 - p_{max}$ is the probability of a failure, and $C_n^x$ represents the possible combinations of $x$ elements out of $n$.

In our case, the probability of an incorrect assignment is equal to the probability of having three failures plus that of having two failures and one success; numerically, this can be represented as $P(\text{incorrect diagnosis}) = C_3^0 0.85^0 \cdot (1 - 0.85)^3 + C_3^1 0.85^1 \cdot (1 - 0.85)^2 \sim 0.001$. This probability is still above the prefixed confidence level (0.005). However, it is lower than the one obtained with a single-point measurement, suggesting that the more trials there are, the lower the probability of failure is. Proceeding by induction, we can easily figure out that, for an odd number of trials $n = 2m + 1$, the probability of an incorrect diagnosis is the cumulative binomial probability $F(x)$ of at most $x = m + 1$ successes, for $m \in N$. We chose to investigate only an odd number of trials, because, for an even number, we cannot assign a unique diagnosis outcome when successes are equal to failures.

In Figure 7, we plot the probability of an incorrect diagnosis as a function of the acquired number of curves calculated using $p_{max}$ from Table 1 for the Sneddon model (blue continuous line) and for the neural network (gold continuous line). Analysis of Figure 7 allows determining the minimum number of FD curves needed to assign the class membership with a 0.005 confidence level. Basically, we chose the minimum number of curves for which the probability to make a mistake was lower than 0.005, which is represented by a dashed horizontal line in Figure 6. The minimum number of curves needed is summarized in Table 1, which confirms that the neural network approach has superior classification ability with respect to the Sneddon model approach, thus allowing us to assign tissue membership with a significantly lower number of measures. Moreover, the automated classification of FD curves with the neural network algorithm has the further advantages of being operator-independent and providing a direct diagnosis, instead of giving a physical parameter that needs to be further interpreted.

Taken together, the data shown in Figure 7 and Table 1 have the potential to dramatically shrink the time needed to classify a tissue according to its pathological state, making AFM competitive with the analysis of conventional histological sections. To give an example, a typical 40× histological section has an area of approximately $0.4 \times 0.3$ mm$^2$; if we consider the problem of healthy vs. tumor tissue, to cover the area mentioned above, approximately 75 maps of $40 \times 40$ microns are required, and, for each, nine curves are required, for a total of $75 \times 9 = 675$ FD curves. Considering also an average piezo excursion of 30 microns, which is used to compensate for the high corrugation of surgically removed tissues, and a piezo scanning speed of 5 μm/s, it is possible to estimate an acquisition time of 4050 s. In this regard, it is worth stressing that, while histology needs extensive tissue preparation, AFM allowed us to measure tissue slices immediately after surgical resection, without any pre-processing step. As a further advantage, data acquired with AFM can be automatically analyzed by our neural network approach, which produces an immediate result in the form of a direct analysis instead of giving physical parameters or images that need to be further interpreted by experienced pathologists or data analysts. The estimated time, which is competitive with the time spent in the pre-processing of histological sections, could be further reduced using high-speed atomic force microscopes, which are currently available on a commercial basis.

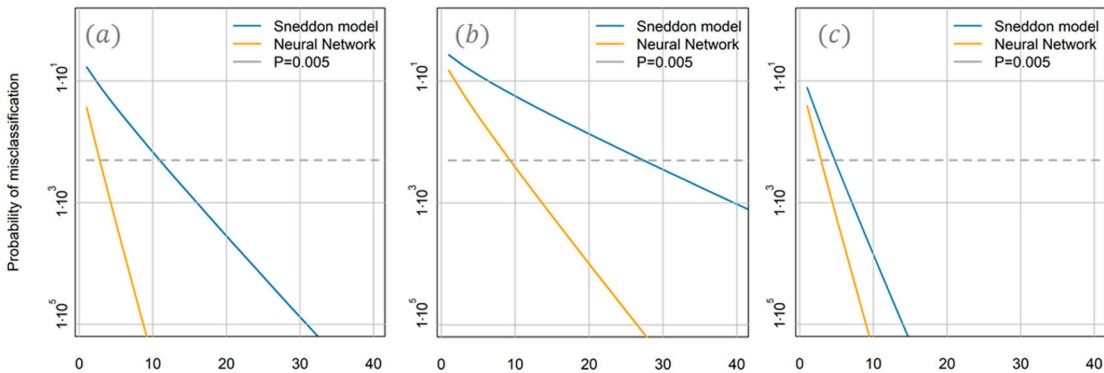

**Figure 7.** Probability plot of an incorrect diagnosis as a function of the acquired number of curves calculated using $p_{max}$ from Table 1 for the Sneddon model (blue continuous line) and for the neural network (gold continuous line). In order, healthy vs. necrotic tissue (**a**), healthy vs. tumor tissues (**b**), and necrotic vs. tumor tissue (**c**) cases are shown. To choose the minimum number of curves needed to perform a correct diagnosis, we fixed the probability to commit an incorrect diagnosis to 0.005 (dashed line).

## 4. Conclusions

Atomic force microscopy in force spectroscopy mode has great potential in cancer diagnostics, being able to distinguish healthy from pathological tissues according to their mechanical properties. However, the translation process of AFM in clinical practice is hindered by the fact that AFM is a time-consuming technique in terms of measurements and analyses. In this study, we attempted to overcome these limitations using two different strategies. With the aim of cutting the time spent analyzing AFM force–distance curves, we tested the possible applicability of neural networks for pattern recognition in the classification of brain cancer tissues. Different tissue types were classified according to the shape of the entire force–distance cycle. The classification performances of the algorithm were evaluated by ROC curves and systematically compared with the performances of the Young's modulus E. Preliminary evidence was reported for the accuracy and precision of the method, and arguments were presented for the preference of neural networks over the conventional methods based on curve-fitting. The former indeed showed comparable or larger AUC values with respect to the latter. This is probably because the Young's modulus E was obtained taking into account only a portion of the approach curve, while the neural network considered the entire approach and retract curves. However, it is important to stress that more complex and comprehensive models are required to demonstrate the general applicability of our conclusions to the field of AFM-based cancer diagnostics; thus, such conclusions are valid only for the particular classification proposed. In order to reduce experimental time, we evaluated the minimum number of FD curves needed to classify a given tissue region with a 0.005 confidence level, in the cases of the neural network and the Young's modulus E. Again, we showed evidence that fewer curves are needed when using the neural network.

Taken together, these results have the potential to promote a shortening of the time spent during measurements and analyses, thus boosting the use of AFM in diagnostics.

**Author Contributions:** Conceptualization, G.C. and T.E.S.; data curation, T.E.S. and V.P.; formal analysis, G.C. and E.M.; investigation, G.C., A.M., T.E.S., M.N., and E.M.; supervision, G.C., M.P., and M.d.S.; writing—review and editing, G.C.

**Funding:** The Italian Ministry of Health ("Progetto Giovani Ricercatori 2014–2015", Grant No. GR-2016-02363310) is gratefully acknowledged.

**Conflicts of Interest:** The authors declare no conflict of interest.

## Appendix A

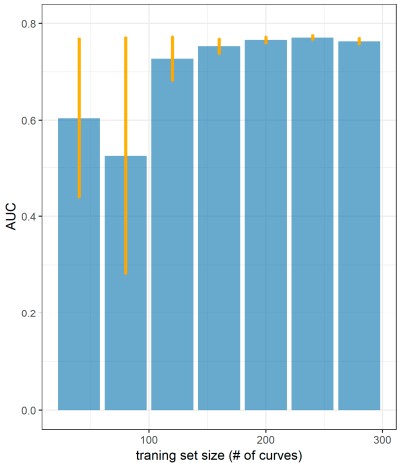

**Figure A1.** AUC (b) of the ROC curves obtained comparing the neural network (NN) outputs with the expected values upon varying the number of the curves in training set. The higher the training set size is, the lower the variability is; for training sets larger than $N = 200$, there is no significant improvement in the algorithm performance. Moreover, at this size, a small variability of the results is observed.

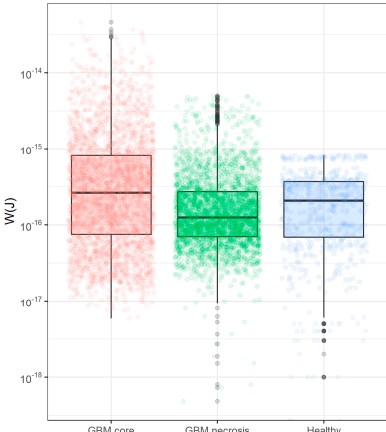

**Figure A2.** Box-plot analysis of the work adhesion values calculated for the three tissues. No statistically significant difference among the three tissue types was found.

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
