# Peer review of "Efficient Spatial Sampling for AFM-Based Cancer Diagnostics: A Comparison between Neural Networks and Conventional Data Analysis"

_condensedmatter, doi:10.3390/condmat4020058_

Round 1

Reviewer 1 Report

Gabriele Ciasca et. al. describe progress in developing neural networks to improve data analysis and acquisition for AFM based cancer diagnostics. I find the manuscript well written and generally interesting. The results, although incremental, provide important advances on their previous work particularly in the ability to distinguish between tumour and healthy tissue. I have the following minor suggestions to improve the manuscript. 

Minor Comments:

For reproducibility it would be useful to state the specific cantilevers used (i.e. model and manufacturer).

The x-range of Figure 1 c) is unnecessarily different to parts a) and b). This makes it difficult to assess the gradient differences qualitatively and misleads readers. Also, the label for panel c) is missing in the figure caption.

Line 188 should read “as shown in Figure 2a” rather than “Figure 1a”.

Caption of Figure 7 is missing labels for a), b) and c).

Author Response

Comments and Suggestions for Authors

Gabriele Ciasca et. al. describe progress in developing neural networks to improve data analysis and acquisition for AFM based cancer diagnostics. I find the manuscript well written and generally interesting. The results, although incremental, provide important advances on their previous work particularly in the ability to distinguish between tumour and healthy tissue. I have the following minor suggestions to improve the manuscript.

We thank the referee for the appreciation of our work.

Minor Comments:

For reproducibility it would be useful to state the specific cantilevers used (i.e. model and manufacturer).

We included this information in the Material and Methods section.

The x-range of Figure 1 c) is unnecessarily different to parts a) and b). This makes it difficult to assess the gradient differences qualitatively and misleads readers. Also, the label for panel c) is missing in the figure caption.

We thank the referee for the careful reading of the paper. The mentioned figure has been modified according to the referee’s suggestion.

Line 188 should read “as shown in Figure 2a” rather than “Figure 1a”.

We modified the sentence according to the referee’s suggestion.

Caption of Figure 7 is missing labels for a), b) and c).

We modified the sentence according to the referee’s suggestion.

Reviewer 2 Report

The simplification of  analysis of mechanical probing data of cancer tissue with diagnostics in mind is definitely a worthy goal. Using neural networks for the analysis of force curves is novel and has a high chance of success. The presented work, however, falls short from providing a compelling data on the effectiveness of the new approach. There are several points that seriously undermine the work and need to be addressed. In short:

How the classification of the tissues from biopsy into "healthy", "necrotic" and "cancer" was performed? At a minimum a proper staining is required for such, however no data on staining, or any other method of classification, are provided.

How many patients and patient samples were analyzed?

How many force curves were collected per sample?

How the conclusion of minimum number of force curves was reached?

What kind of AFM instrument was used? What brand of probes?  

What is the "biocompatible glue"?

Adhesion forces between sample and the tip should be taken into consideration.

Application of indentation force (only?) of 5nN seems on the hight end and left without justification.

The resolution of elasticity maps is missing.

Selection of 200 FDs seems arbitrary - any justification for this number?

Author Response

Thank yo uvery much for your valuable comments, here are our response to your comments:

Reviewer 3 Report

The manuscript reads well and can be publish after a minor revision.

Few comments:

-line 21: define ROC

-There are some overstatements: e.g. Line 14- "AFM has ability to distinguish between healthy and cancer tissues". At this time, I would tone down a bit this statement.

Line 70 " ...operator-independent method". The AFM data acquisition depends on the operator a lot. This statement has be toned down too.

-Line 271: "...shorten AFM acquisition time." -overstatement. 

Author Response

The manuscript reads well and can be publish after a minor revision.

We thank the referee for the appreciation of our work.

Few comments:

-line 21: define ROC

Line 21 was modified as follows:

The performance of the classifier is evaluated with receiving operating characteristic (ROC) curves for the approach and retract curves separately and in combination with each other.”

-There are some overstatements: e.g. Line 14- "AFM has ability to distinguish between healthy and cancer tissues". At this time, I would tone down a bit this statement.

We agree with the referee that the quoted sentence is too much generic. As a matter of fact, AFM is able to discriminate between different classes of tissues under very specific experimental conditions. Therefore, we lowered our statement as follows:

“Atomic Force Microscopy (AFM) in spectroscopy mode has received a lot of attention because of its potential in distinguishing between healthy and cancer tissues”

Line 70 " ...operator-independent method". The AFM data acquisition depends on the operator a lot. This statement has be toned down too.

We removed the quoted statement, rearranging the sentence as follows:

It is worth stressing that AFM has also several advantages over these methodologies, because this technique yields quantitative results which can be statistically analyzed”

-Line 271: "...shorten AFM acquisition time." -overstatement.

The quoted sentence was modified as follows:

“In this section, we discuss a possible strategy to evaluate the minimum number of FD curves needed to properly classify a given scanning area, with a predetermined level of accuracy”

Round 2

Reviewer 2 Report

The concerns have been adequately addressed -thank you!